# Organic Acid-Catalyzed Subcritical Water Hydrolysis of Immature *Citrus unshiu* Pomace

**DOI:** 10.3390/foods11010018

**Published:** 2021-12-22

**Authors:** Sang-Bin Lim

**Affiliations:** Department of Food Bioengineering, Jeju National University, Jeju 63243, Korea; sblim@jejunu.ac.kr; Tel.: +82-64-754-3617

**Keywords:** immature *Citrus unshiu* pomace, flavonoids, subcritical water hydrolysis, response surface methodology, enzyme inhibitory activity

## Abstract

Immature *Citrus unshiu* pomace (ICUP) was hydrolyzed under organic acid-catalyzed, subcritical water (SW) conditions to produce flavonoid monoglucosides (hesperetin-7-*O*-glycoside and prunin) and aglycons (hesperetin and naringenin) with high biological activities. The results of single-factor experiments showed that with 8 h of hydrolysis and an increasing citric acid concentration, the yield of flavonoid monoglucosides (hesperetin-7-*O*-glycoside and prunin) increased from 0 to 7% citric acid. Afterward, the hesperetin-7-*O*-glycoside yield remained constant (from 7 to 19% citric acid) while the pruning yield decreased with 19% of citric acid, whereas the aglycon yield increased continuously. In response surface methodology analysis, a citric acid concentration and hydrolysis duration of 13.34% and 7.94 h were predicted to produce the highest monoglucoside yield of 15.41 mg/g, while 18.48% citric acid and a 9.65 h hydrolysis duration produced the highest aglycon yield of 10.00 mg/g. The inhibitory activities of the SW hydrolysates against pancreatic lipase (PL) and xanthine oxidase (XO) were greatly affected by citric acid concentration and hydrolysis duration, respectively. PL and α-glucosidase inhibition rates of 88.2% and 62.7%, respectively, were achieved with 18.48% citric acid and an 8 h hydrolysis duration, compared to 72.8% for XO with 16% citric acid and 12 h of hydrolysis. This study confirms the potential of citric acid-catalyzed SW hydrolysis of ICUP for producing flavonoid monoglucosides and aglycons with enhanced enzyme inhibitory activities.

## 1. Introduction

Citrus fruits are widely consumed worldwide as juices. However, after juice extraction, a large amount of pomace can be produced that includes peels, pulp, and seeds. Citrus pomace is a major potential source of flavonoid glucosides [1]. Many of these compounds have beneficial biological effects, including antioxidant, antimicrobial, anti-obesity, antihyperglycemic, anti-inflammatory, and anticancer effects [2,3,4]. In particular, immature citrus fruits yield pomace with higher amounts of flavanones than mature fruits, and are thus potentially rich sources of flavonoids [5].

Most flavonoids are bound to sugars by glycosidic bonds in plants, and their biological activities depend on their glycosylation patterns and structural features [6]. Several studies have reported that the biological activities and bioavailability of flavonoid glucosides increase after deglycosylation; therefore, flavonoid monoglucosides and aglycons may have greater biological activities than their di-glycosylated flavonoids, and thus high potential for application in the food and pharmaceutical industries [7,8,9].

Cleavage of glycosidic bonds in flavonoid diglycosides, to release monoglucosides and aglycons, requires chemical or enzymatic hydrolysis [10,11,12,13]. Chemical hydrolysis is a nonselective process, the harsh conditions of which can damage the diglycoside structure [14,15]. Enzymatic hydrolysis often occurs under mild conditions with high selectivity, but requires a long reaction time that limits industrial applications [16,17,18]. Commercially available L-rhamnosidases and D-glucosidases are excellent enzymes for the hydrolysis of flavonoid rutinosides [19,20,21].

Subcritical water (SW), a green solvent that maintains a liquid state at 100–374 °C under pressure, has been used for extracting compounds from agricultural and food processing wastes. Recently, SW hydrolysis has been recognized as a promising alternative to chemical and enzymatic hydrolysis because of its environmental friendliness. SW can be used to hydrolyze flavonoid diglycosides into monoglucosides and aglycons. This is because, in the subcritical region, the ionic product of water increases with temperature and an acidic medium favorable to hydrolysis reactions is formed [22,23,24]. Ruen-ngam et al. [25] performed SW hydrolysis of hesperidin over 1–4 h at 110–140 °C, with a CO_2_ pressure of 25 MPa. However, the high pressure limits industrial applications. In our previous study, SW was used to extract flavonoids from *Citrus unshiu* peels at 145–175 °C, and flavonoid monoglucosides (hesperetin-7-*O*-glucoside and prunin) and aglycons (hesperetin and naringenin) were produced at temperatures exceeding 165 °C [8]. However, under high-temperature SW, side reactions or elution of unnecessary compounds and non-polar components may occur, and industrial application is limited because the high temperature may damage equipment [22,24,26]. Valdivieso Ramirez et al. [27] also reported that the high-temperature processing of agricultural residues with complex substrates produced oligosaccharides and unwanted side reaction products, which might interfere with downstream purification processes. Thus, despite the potential of SW for hydrolytic reaction catalysis, the high temperature requirement remains a challenge.

Through catalysis with organic acids, lower-temperature (120 °C) SW hydrolysis can be realized. Several organic acid-based processes have been explored thus far, such as pressurized extraction of anthocyanins and flavonols from grape pomace using a hydroethanolic solvent acidified with organic acids (formic, acetic, citric, and tartaric acids) [28]; deglycosylation of glycosylated quercetins from apple pomace using deep eutectic solvents with carboxylic acids [9]; and production of oligosaccharides from pea hull fiber using SW with carboxylic acids [27]. The addition of carboxylic acids to SW can control the degradation of arabinose in aqueous maleic and fumaric acids under high-temperature conditions [27,29]. However, organic acid-catalyzed SW hydrolysis has not been investigated for the low-temperature conversion of flavonoid diglycosides from immature citrus pomace into high-biological-activity monoglucosides and aglycons.

The objective of this study was to produce flavonoid monoglucosides and aglycons from immature *C. unshiu* pomace (ICUP) under citric acid-catalyzed, low-temperature (120 °C) SW conditions to make functional foods targeting for weight control. First, the effects of citric acid concentration and hydrolysis duration on monoglucoside and aglycon yields were evaluated through single-factor experiments. Afterward, the hydrolysis conditions were optimized via response surface methodology (RSM) to maximize the hydrolysis yields. The enzyme-inhibiting effects of SW hydrolysates against pancreatic lipase (PL), α-glucosidase (αG), and xanthine oxidase (XO) were evaluated.

## 2. Materials and Methods

### 2.1. Sample Preparation

Immature *C*. *unshiu* fruits were obtained from a farm in Jeju, Korea. The fruits were rinsed, quartered, and blended for 90 s. The homogenates were pressed with a screw-type extractor. The pomace was separated from the juice, freeze-dried, crushed into powder (14–50 mesh), and stored in a freezer.

### 2.2. Chemicals

Hesperidin, hesperetin, narirutin, naringenin, xanthine oxidase from bovine milk, xanthine, allopurinol, ρ-nitrophenyl-β-D-glucopyranoside (ρ-NPG), acarbose, α-glucosidase from Saccharomyces cerevisiae, ρ-nitrophenyl butyrate (ρ-NPB), lipase from porcine pancreas, orlistat, and 3-(*N*-Morpholino)propane sulfonic acid (MOPS) were purchased from Sigma Chemical Co. (St. Louis, MO, USA). Hesperetin-7-*O*-glucoside and prunin were purchased from Extrasynthese (Genay, France). Citric acid was purchased from Junsei Chemical Co., Ltd. (Chuo-ku, Tokyo, Japan) and HPLC grade acetonitrile and methyl alcohol were purchased from Daejung Chemicals & Metals Co., Ltd. (Shiheung, Gyeonggi, Korea).

### 2.3. Hydrolysis of Acidified SW

Samples consisting of 1 g ICUP powder were mixed with a certain amount of citric acid and 30 mL distilled water and loaded into an autoclave (Jeio Tech, Co., Ltd., Daejeon, Korea). The mixtures were heated to 120 °C for the desired hydrolysis duration. After hydrolysis, the mixtures were filtered with a Toyo No. 5A filter paper, and distilled water was added to the mixtures to adjust their volume to 50 mL. The mixtures were then filtered with a 0.45 μm syringe filter, and the SW hydrolysates were mixed with methanol at a 1:1 (*v*/*v*) ratio and used for individual flavonoid analyses and enzyme inhibitory activity analyses.

The total hydrolysis yields (%) of flavonoid monoglucosides and aglycons were calculated according to the following equation:(1)$$\mathrm{Total}\;\mathrm{hydrolysis}\;\mathrm{yield}\;(\%)=\frac{Q_{H}\times \left({\mathrm{MW}}_{2G}/{\mathrm{MW}}_{H}\right)}{Q_{2G}}\times 100$$
where Q_H_ is the quantity of each hydrolysis product in SW hydrolysates (μg/g dry sample); Q_2G_ is the quantity of flavonoid diglycosides in the raw sample (μg/g dry sample); and MW_2G_ and MW_H_ are the molecular weights of diglycosides and each hydrolysis product, respectively.

### 2.4. Single-Factor Experiments

The effects of hydrolysis parameters (citric acid concentration and hydrolysis duration) on the yields of flavonoid monoglucosides and aglycons from ICUP were investigated. The effects of the citric acid amount added to the dry ICUP sample were evaluated at 4, 7, 10, 13, 16, and 19% (wt/wt) after 8 h of hydrolysis, with a temperature of 120 °C. The effects of hydrolysis duration were evaluated after 4, 6, 8, 10, 12, 14, and 16 h, also with a temperature of 120 °C and with 10% of citric acid.

### 2.5. Response Surface Methodology

A central composite design was used in the experiment, which was conducted at a temperature of 120 °C and had two independent variables with five levels (citric acid concentration [X_1_]: 1.51, 4, 10, 16, and 18.48%; hydrolysis duration [X_2_]: 2.35, 4, 8, 12, and 13.65 h). There were eight runs with independent variables (runs 1–8), and five replicates at the central point (runs 9–13). A second-order polynomial equation was used to investigate the relationship between the independent variables (citric acid concentration and hydrolysis duration) and responses (monoglucoside and aglycon yields, total hydrolysis products, % total hydrolysis yield).

### 2.6. High-Performance Liquid Chromatography

The hydrolysate contents of SW, including hesperidin, narirutin, hesperetin-7-*O*-glucoside, prunin, hesperetin, and naringenin, were quantified with an Alliance 2965 high-performance liquid chromatography system (Waters Corp., Milford, MA, USA) [8]. An Inertsil ODS-3V column (4.6 mm × 250 mm, 5 μm particle size, GL Science, Tokyo, Japan) was used to separate each flavonoid from the hydrolysate. The mobile phase consisted of 0.5% acetic acid solution (phase A) and acetonitrile (phase B), and the flow rate was 1.0 mL/min (gradient of B = 15% at 0 min, 25% at 8 min, 25% at 15 min, 65% at 32 min, and 15% at 33 min). The column temperature, injection volume, and equilibrium time were 35 °C, 10 μL, and 2 min, respectively. Each flavonoid was detected at 290 nm. To identify and quantify the flavonoids, their retention times, UV–visible spectra, and peak areas were compared to those of the standard compounds.

### 2.7. Enzyme Inhibitory Activity Analysis

The inhibitory activities of SW hydrolysates against PL, αG, and XO were measured. Methanol-diluted hydrolysates (two-fold dilution) were used for enzyme inhibitory activity analysis. The PL inhibitory activity was measured according to the method established by Kim et al. [30]. A 0.05 mL sample was mixed with 0.025 mL PL enzyme solution (1 mg/mL in (*N*-morpholino) propanesulfonic acid sodium salt (MOPS) buffer (10 mM MOPS and 1 mM ethylenediaminetetraacetic acid, pH 6.8)) and 0.05 mL tris HCl buffer (100 mM tris HCl and 5 mM CaCl_2_, pH 7.0). The mixture was incubated at 37 °C for 15 min; then, 0.05 mL of 10 mM 4-nitrophenyl butyrate was added, and the obtained mixture was re-incubated at 37 °C for 30 min. The absorbance at 405 nm was measured.

The αG inhibitory activity was measured according to the method of Proença et al. [31]. A 0.025 mL sample was mixed with 0.025 mL of αG solution (1 U/mL in 100 mM phosphate buffer, pH 6.8) and 0.1 mL phosphate buffer (100 mM, pH 6.8). This mixture was incubated at 25 °C for 15 min. Then, 0.05 mL of 3 mM 4-nitrophenyl β-D-glucopyranoside was added, and the obtained mixture was re-incubated at 25 °C for 5 min. The absorbance at 405 nm was measured every 4 min, for a total of 10 measurements. The velocities of the reactions before and after enzyme addition were calculated.

The XO inhibitory activity was measured according to the method of Yoon et al. [32]. A 0.025 mL sample was mixed with 0.025 mL XO solution (0.5 U/mL; in 50 mM phosphate buffer, pH 7.5) and 0.1 mL phosphate buffer (50 mM, pH 7.5). This mixture was incubated at 25 °C for 15 min. Then, 0.05 mL of 2 mM xanthine solution was added, and the obtained mixture was incubated at 25 °C for 5 min. The absorbance at 290 nm was measured every 4 min, for a total of 10 measurements. The velocities of the reactions before and after enzyme addition were calculated.

For all enzyme inhibitory activity measurements, 50% methanol was used as the control.

### 2.8. Statistical Analyses

The experimental data were analyzed using Duncan’s multiple range test (*p* < 0.05), performed with Statistical Package for the Social Sciences (SPSS) software (ver. 24.0; SPSS Inc., Chicago, IL, USA). SW hydrolysis was optimized through regression analysis performed with Minitab (ver. 18.1; Minitab Inc., State College, PA, USA). Pearson correlation coefficients between enzyme inhibitory activities and the flavonoid contents of SW hydrolysates were also calculated with Minitab

## 3. Results and Discussion

### 3.1. Single-Factor Effects Experiments

The effects of SW hydrolysis parameters (citric acid concentration and hydrolysis duration) on the yields of flavonoid monoglucosides and aglycons from ICUP were evaluated. Figure 1 illustrates the effects of citric acid concentration (4–19% (wt/wt)) on the hydrolysis yields of flavonoid monoglucosides and aglycons, at a hydrolysis temperature and duration of 120 °C and 8 h, respectively. With increasing citric acid concentration, the yield of flavonoid monoglucosides (hesperetin-7-*O*-glycoside and prunin) increased from 0 to 7% citric acid. Afterward, the hesperetin-7-*O*-glycoside yield remained constant (from 7 to 19% citric acid) while the pruning yield decreased with 19% of citric acid. This trend is attributable to the conversion of flavonoid monoglucosides into aglycons at high citric acid concentrations. The yields of flavonoid aglycons (hesperetin and naringenin) increased with increasing citric acid concentration from 0 to 19%, because monoglucosides conversion into aglycons was further enhanced at high citric acid concentrations. Generally, the hydrolysis yields of flavonoids increased with increasing citric acid concentration, owing to a decrease in pH [9,28].

The effects of hydrolysis duration (4–16 h) on monoglucoside and aglycon yields were measured at 120 °C with 10% citric acid (Figure 2). The yields of hesperetin-7-*O*-glycoside and prunin increased with increasing hydrolysis duration from 4 to 8 h, but decreased thereafter. The yields of hesperetin and naringenin also increased with increasing hydrolysis duration from 4 to 12 h, but decreased thereafter. This is attributable to the decomposition and further degradation of flavonoids into low-molecular-weight compounds with prolonged exposure to an acidic medium at high temperatures [27,33]. These results indicate that deglycosylation of citrus flavanones from ICUP can be controlled by varying the citric acid concentration in SW and hydrolysis duration.

In preliminary organic acid-catalyzed SW hydrolysis experiments conducted at 120 °C for 10% organic acids and 8 h of hydrolysis (Appendix A), citric acid (pKa 3.14) produced the highest hydrolysis yields of flavonoid monoglucosides and aglycons from ICUP, followed by tartaric (pKa 2.98), formic (3.75), malic (3.40), ascorbic (4.19), succinic (4.16), and acetic acids (4.76). The differences in yields were probably related to difference in acidity (pKa values) among the organic acids [9,28]. Yu and Bulone [9] also reported that the acidity of deep eutectic solvents enhanced de-glycosylation of quercetin derivatives during extraction from apple pomace.

### 3.2. Response Surface Optimization

The hydrolysis parameters (citric acid concentration and hydrolysis duration) were optimized through a central composite design with RSM (Table 1), to obtain the highest yields of flavonoid monoglucosides and aglycons from ICUP. The hydrolysis yield ranged from 6.47 to 12.09 mg/g sample for hesperetin-7-*O*-glucoside, 1.99 to 8.05 mg/g sample for hesperetin, 2.05 to 3.45 mg/g sample for prunin, 0.53 to 2.20 mg/g sample for naringenin, 8.52 to 15.54 mg/g sample for monoglucosides (H7G + PR), 2.52 to 10.25 mg/g sample for aglycons (HT + NG), and 11.44 to 24.60 mg/g sample for total hydrolysis products (H7G + PR + HT + NG). The total hydrolysis yields of flavonoid monoglucosides and aglycons were 22.9–53.1% at citric acid concentrations of 1.51–18.48% and hydrolysis durations of 2.35–13.65 h.

Ko et al. [34] reported that aglycons such as hesperetin from lemons (about 25 mg/g sample at 190 °C) and naringenin from grapefruits (about 2.37 mg/g sample at 170 °C) were extracted at higher temperatures than glycoside such as hesperidin and narirutin because glycosides are unstable at high temperatures and long extraction times in the SW extraction and degraded to low-molecular weight aglycons at temperatures > 170 °C due to their thermal instability. Ko et al. [35] indicated that naringenin (about 7.2 mg/g sample at 190 °C) could be extracted at higher temperatures (≥170 °C) than hesperidin and narirutin on pilot-scale SW extraction from *C. unshiu* peel. Kim and Lim [8] found that the SW extract from *C. unshiu* peel at 175 °C was high in flavonoid monoglucosides (hesperetin-7-*O*-glucoside 2.04 and prunin 0.31 mg/g sample) and aglycons (hesperetin 1.80 and naringenin 0.26 mg/g sample).

Table 2 presents the data obtained from analysis of variance of the regression coefficients of citric acid-catalyzed SW hydrolysis. The regression models for monoglucoside and aglycon yields, total hydrolysis products, and total hydrolysis yield (%) fit well with the experimental data, with low *p*-values (*p* < 0.05) and high *R*^2^ values (*R*^2^ ≥ 0.9761). The following second-order polynomial models provided the optimal conditions to maximize flavonoid hydrolysis yields:(2)$$Y\;(\mathrm{Monoglucosides})=15.02+1{.39X}_{1}-\;0{.13X}_{2}-\;1{.22X}_{1}{}^{2}-\;2{.57X}_{2}{}^{2}-2.18X_{1}X_{2}$$
(3)$$Y\;(\mathrm{Aglycons})=8.02+1{.60X}_{1}+1{.46X}_{2}-\;0{.29X}_{1}{}^{2}-\;1{.78X}_{2}{}^{2}-0.13X_{1}X_{2}$$
(4)$$Y\;(\mathrm{Total}\;\mathrm{hydrolysis}\;\mathrm{products})=23.04+2{.99X}_{1}+1{.33X}_{2}-\;1{.52X}_{1}{}^{2}-\;4{.37X}_{2}{}^{2}-2.13X_{1}X_{2}$$
(5)$$Y\;(\%\;\mathrm{Total}\;\mathrm{hydrolysis}\;\mathrm{yield})=48.24+6{.81X}_{1}+3{.78X}_{2}-\;2{.96X}_{1}{}^{2}-\;9{.38X}_{2}{}^{2}-4.20X_{1}X_{2}$$
where X_1_ and X_2_ are the test variables (citric acid concentration and hydrolysis duration, respectively).

The regression coefficient for each term in Table 2 indicates the effects of the two variables (X_1_ and X_2_) on the hydrolysis yields. The response surface plots visually illustrate the effects of each variable on the hydrolysis yields (Appendix A).

The models for total hydrolysis products and total hydrolysis yield (%) show significant positive coefficients for the linear terms (X_1_ and X_2_) and significant negative coefficients for the quadratic terms (X_1_^2^ and X_2_^2^) (all *p* < 0.05). The results indicate that with increasing citric acid concentration and hydrolysis duration, the hydrolysis yields increase, reach a peak, and then decrease. Moreover, the interaction term between citric acid concentration and hydrolysis duration (X_1_X_2_) is also significant (*p* < 0.05), indicating that the interaction between X_1_ and X_2_ significantly affects the yield of total hydrolysis products. In the model for monoglucoside yield, all of the terms, excluding the linear term X_2_, are significant (*p* < 0.05), while for flavonoid aglycon yield, only the linear terms (X_1_ and X_2_) and quadratic term (X_2_^2^) are significant (*p* < 0.05).

According to the optimized regression models, the optimal hydrolysis conditions for monoglucoside yield are a citric acid concentration and hydrolysis duration of 13.34% and 7.94 h, respectively (predicted yield = 15.41 mg/g sample). The optimal hydrolysis conditions for aglycon yield are a citric acid concentration and hydrolysis duration of 18.48% and 9.65 h, respectively (predicted yield = 10.0 mg/g sample) (Table 3).

The optimum citric acid concentration and hydrolysis duration for aglycon production are 5.14% and 1.71 h higher than those for monoglucoside production, because flavonoid monoglucosides are further converted into aglycons with higher citric acid concentrations and longer hydrolysis durations. The optimal hydrolysis conditions for the simultaneous production of flavonoid monoglucosides and aglycons are a citric acid concentration of 15.91% and hydrolysis duration of 8.62 h (predicted maximum yield = 24.62 mg/g sample).

### 3.3. Enzyme Inhibitory Activities

The enzyme inhibitory activities of SW hydrolysates against PL, αG, and XO were measured (Table 4). These activities are related to weight control [30], anti-diabetes effects [31], and anti-gout effects [32], respectively. The citric acid concentration in the methanol-diluted hydrolysates (two-fold dilution) used for enzyme inhibitory activity analysis was <1% (wt/v), and did not affect enzyme activities.

The inhibitory activity of SW hydrolysates against PL was greatly affected by citric acid concentration, while that against XO was greatly affected by hydrolysis duration. With increasing citric acid concentration (at a fixed hydrolysis duration), PL was most inhibited, followed by αG, and then XO. Under hydrolysis durations of 4 and 12 h, with increasing citric acid concentration from 4 to 16%, the inhibitory activity of the hydrolysates against PL increased 2.2-fold (from 37.0 to 81.4%) and 1.8-fold (from 45.7 to 83.6%), respectively. In contrast, the inhibitory activity against XO increased only 1.1-fold (from 27.1 to 28.7%) and 1.3-fold (from 55.7 to 72.8%), respectively.

With increasing hydrolysis duration (at a fixed citric acid concentration), XO was most inhibited, followed by αG and PL. Under citric acid concentrations of 4 and 16%, with increasing hydrolysis duration from 4 to 12 h, the inhibitory activity of the hydrolysates against XO increased 2.0-fold (from 27.1 to 55.7%) and 2.5-fold (from 28.7 to 72.8%), respectively. In contrast, the inhibitory activity against PL increased only 1.2-fold (from 37.0 to 45.7%) and 1.0-fold (81.4 to 83.6%), respectively. The highest inhibitory activities against PL and αG were 88.2 and 62.7%, respectively, at a citric acid concentration and hydrolysis duration of 18.48% and 8 h, respectively, while the highest inhibitory activity against XO was 72.8% with a 16% citric acid concentration and 12 h hydrolysis duration. The inhibitory activity of the SW hydrolysates against acetylcholinesterase, which is related to the promotion of memory formation and consolidation, was also measured. No inhibitory effects were found.

Kim and Lim [8] measured the inhibitory activities of SW extracts from *C. unshiu* peel and found that the extracts at 175 °C with high in flavonoid monoglucosides and aglycons exhibited inhibitory activities of 86.8, 55.5, and 52.6% against PL, αG, and XO, respectively. Cvetanovic et al. [36] reported that SW extract from *Sambucus ebulus* leaves was high in total phenolics (116.3 mg CAE/g extract) and exhibited high αG inhibiting effects (2.04 mmol acarbose equivalents/g extract).

### 3.4. Correlations between Enzyme Inhibitory Activities and Flavonoid Yields

To determine the flavonoids affecting the enzyme inhibitory activities of the SW hydrolysates, Pearson correlation coefficients describing the relationships between functional properties and flavonoid yields were calculated (Table 5).

The enzyme inhibitory activities of the SW hydrolysates were strongly correlated with the flavonoid yields, i.e., PL inhibition versus naringenin (*r* = 0.721), total hydrolysis products (*r* = 0.691), aglycons (HT + NG) (*r* = 0.686), and hesperetin (*r* = 0.675); αG inhibition versus hesperetin (*r* = 0.973), aglycons (HT + NG) (*r* = 0.973), and naringenin (*r* = 0.966); and XO inhibition versus naringenin (*r* = 0.777), aglycons (HT + NG) (*r* = 0.750), and hesperetin (*r* = 0.742). The yield of aglycons (naringenin and hesperetin) was closely correlated with the enzyme inhibitory activities of PL, αG, and XO, because aglycone yield varied more widely with citric acid concentration and hydrolysis duration compared to monoglucoside yield (low *r* values).

## 4. Conclusions

Flavonoid monoglucoside and aglycon-rich SW hydrolysates with high enzyme inhibitory activities were produced through the hydrolysis of ICUP under citric acid-catalyzed SW (120 °C) conditions. This hydrolysis process should be cost-effective due to the use of citric acid and relatively low SW temperature. The optimized conditions obtained in this study can be used for large-scale autoclave-based hydrolysis of citrus pomace, facilitating agriculture waste utilization. Flavonoid monoglucosides and aglycons-rich hydrolysates could be used as a functional ingredient in the functional food and nutraceutical industries. Further works are needed on the validation of the predicted values by the experimental values at the optimal hydrolysis conditions for the production of flavonoid monoglucosides and aglycons, and the measurement of enzyme inhibitory activities of the hydrolysates.

## Figures and Tables

**Figure 1 foods-11-00018-f001:**
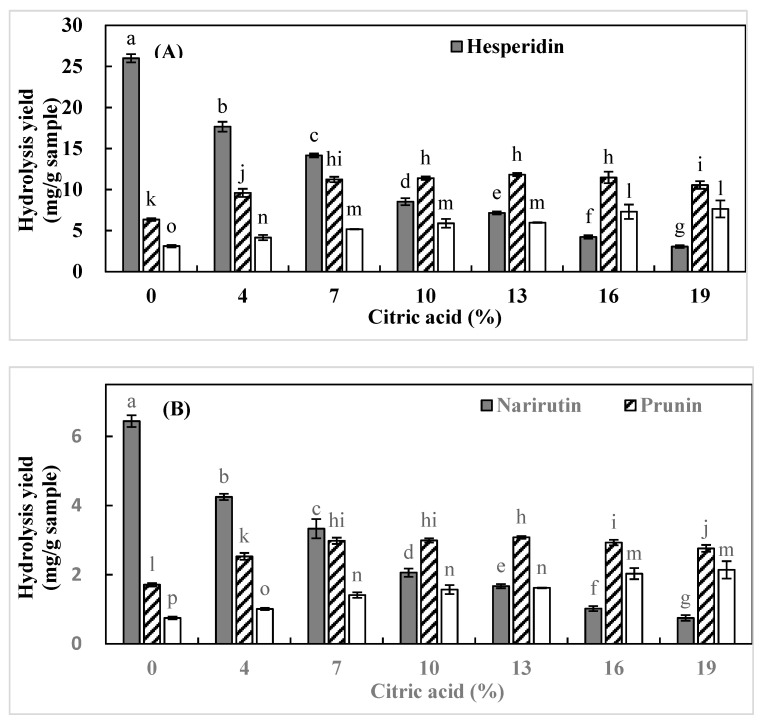
Hydrolysis yields of (**A**) hesperetin-7-*O*-glucoside and hesperetin and (**B**) prunin and naringenin obtained from immature *Citrus unshiu* pomace under different citric acid concentrations, a temperature of 120 °C, and a hydrolysis duration of 8 h. Bars accompanied by different letters (a–o) indicate significant differences (*n* = 3) (*p* < 0.05, Duncan’s test).

**Figure 2 foods-11-00018-f002:**
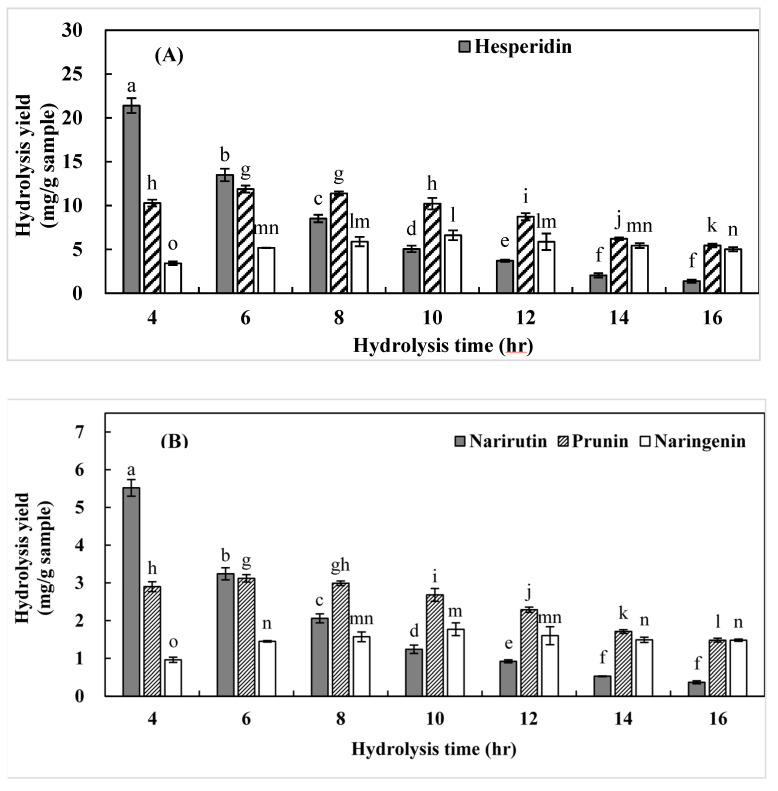
Hydrolysis yields of (**A**) hesperetin-7-*O*-glucoside and hesperetin and (**B**) prunin and naringenin obtained from immature *Citrus unshiu* pomace under different hydrolysis durations, with a temperature of 120 °C, and 10% citric acid. Bars accompanied by different letters (a–o) indicate significant differences (*n* = 3) (*p* < 0.05, Duncan’s test).

**Table 1 foods-11-00018-t001:** Central composite design and corresponding flavonoid yields in subcritical water hydrolysates from immature *Citrus unshiu* pomace.

Run No.	Uncoded (Coded) Levels		Hydrolysis Yield (mg/g Sample)									Total Hydrolysis Yield (%)
	Citric Acid Concentration (X_1_, %)	Hydrolysis Time (X_2_, h)	HD	H7G	HT	NT	PR	NG	H7G + PR	HT + NG	H7G + PR + HT + NG	
1	4 (−1)	4 (−1)	26.87 ± 0.20	6.47 ± 0.06	2.32 ± 0.04	7.70 ± 0.00	2.05 ± 0.00	0.60 ± 0.00	8.52	2.92	11.44	22.9
2	4 (−1)	12 (+1)	16.60 ± 0.41	12.09 ± 0.21	4.48 ± 0.00	4.33 ± 0.10	3.45 ± 0.04	1.28 ± 0.02	11.80	6.18	17.98	37.6
3	16 (+1)	4 (−1)	10.34 ± 0.09	9.33 ± 0.05	4.83 ± 0.26	2.72 ± 0.10	2.47 ± 0.01	1.35 ± 0.07	15.54	5.76	21.31	43.0
4	16 (+1)	12 (+1)	1.68 ± 0.13	7.99 ± 0.09	6.59 ± 0.03	0.38 ± 0.03	2.11 ± 0.00	1.91 ± 0.00	10.10	8.50	18.60	40.9
5	1.51(−1.414)	8 (0)	25.13 ± 0.26	8.10 ± 0.15	3.87 ± 0.01	6.23 ± 0.31	2.13 ± 0.04	0.96 ± 0.00	10.23	4.83	15.07	31.1
6	18.48(+1.414)	8 (0)	4.19 ± 0.32	11.40 ± 0.00	8.05 ± 0.02	0.99 ± 0.02	2.95 ± 0.01	2.20 ± 0.01	14.35	10.25	24.60	53.1
7	10 (0)	2.35(−1.414)	24.08 ± 0.14	6.88 ± 0.11	1.99 ± 0.12	7.55 ± 0.21	2.32 ± 0.04	0.53 ± 0.02	9.20	2.52	11.71	23.0
8	10 (0)	13.65(+1.414)	2.63 ± 0.27	7.86 ± 0.49	5.11 ± 0.69	0.69 ± 0.12	2.11 ± 0.11	1.48 ± 0.14	9.97	6.59	16.56	35.5
9	10 (0)	8 (0)	10.38	12.07	6.59	2.45	3.13	1.69	15.20	8.28	23.48	49.2
10	10 (0)	8 (0)	10.23	11.86	6.38	2.47	3.14	1.67	15.00	8.05	23.05	48.3
11	10 (0)	8 (0)	10.23	11.93	6.33	2.39	3.15	1.67	15.08	8.00	23.08	48.3
12	10 (0)	8 (0)	8.89	12.01	5.46	2.38	3.12	1.67	15.13	8.13	23.26	48.7
13	10 (0)	8 (0)	8.94	11.64	6.08	2.17	3.05	1.60	14.69	7.68	22.37	46.7

Data (runs 1–8) are expressed as the mean ± standard deviation of duplicate experiments. HD: hesperidin, H7G: hesperetin-7-*O*-glycoside, HT: hesperetin, NT: narirutin, PR: prunin, NG: naringenin. H7G + PR: monoglucosides, HT + NG: aglycons, H7G + PR+ HT + NG: total hydrolysis products.

**Table 2 foods-11-00018-t002:** Analysis of variance results for the regression models.

	Monoglucosides (H7G + PR)		Aglycons (HT + NG)		Total Hydrolysis Products		Total Hydrolysis Yields (%)	
Source	F Value	*p*-value	F Value	*p*-Value	F Value	*p*-Value	F Value	*p*-Value
Model	57.19	0.000	78.43	0.000	108.05	0.000	114.76	0.000
X_1_	51.29	0.000	133.97	0.000	157.26	0.000	178.93	0.000
X_2_	0.47	0.513	112.57	0.000	31.28	0.001	55.20	0.000
X_1_^2^	34.58	0.001	3.97	0.087	35.18	0.001	29.43	0.001
X_2_^2^	152.86	0.000	144.98	0.000	290.90	0.000	295.38	0.000
X_1_X_2_	62.78	0.000	0.44	0.528	46.85	0.000	33.99	0.000
Lack of fit	16.62	0.010	5.96	0.059	4.82	0.081	4.18	0.100
R^2^	0.9761		0.9825		0.9872		0.9879	
Pred R^2^	0.8399		0.8931		0.9244		0.9305	
Adj R^2^	0.9590		0.9699		0.9781		0.9793	

X_1_: citric acid concentration (%), X_2_: hydrolysis time (h), pred *R*^2^: predicted *R*^2^, adj *R*^2^: adjusted *R*^2^. H7G: hesperetin-7-*O*-glycoside, PR: prunin, HT: hesperetin, NG: naringenin. Total hydrolysis products: H7G + PR + HT + NG.

**Table 3 foods-11-00018-t003:** Optimized hydrolysis conditions and predicted yields of flavonoid monoglucosides and aglycons.

	Goal	Citric Acid Concentrations (%)	Hydrolysis Time (h)	Predicted Yield (mg/g Sample)	Desirability
H7G + PR	Maximun	13.34	7.94	15.41	0.9824
HT + NG	Maximun	18.48	9.65	10.00	0.9683
H7G + PR + HT + NG	Maximun	15.91	8.62	24.62	1
Total hydrolysis yield (%)	Maximun	16.94	8.85	52.53	0.9813

H7G + PR: hesperetin-7-*O*-glycoside + prunin (monoglucosides), HT + NG: hesperetin + naringenin (aglycons), H7G + PR + HT + NG: total hydrolysis products.

**Table 4 foods-11-00018-t004:** Enzyme inhibitory activities of subcritical water hydrolysates from immature *Citrus unshiu* pomace.

Citric Acid Concentration (%)	Hydrolysis Time (h)	Pancreatic Lipase (%)	α-Glucosidase (%)	Xanthine Oxidase (%)
1.51	8	33.8 ± 1.6 f	40.1 ± 1.7 e	42.2 ± 7.2 c
4	4	37.0 ± 1.6 f	23.3 ± 2.6 f	27.1 ± 3.0 d
4	12	45.7 ± 2.0 e	40.8 ± 0.8 e	55.7 ± 2.1 b
10	2.35	60.9 ± 1.4 d	19.5 ± 1.9 g	21.5 ± 2.4 d
10	13.65	65.7 ± 2.1 c	41.5 ± 3.9 de	69.8 ± 3.3 a
10	8	64.7 ± 3.3 cd	48.6 ± 0.8 c	57.0 ± 4.7 b
10	8	66.9 ± 3.9 c	48.5 ± 2.1 c	56.6 ± 0.4 b
10	8	65.2 ± 3.1 c	45.9 ± 1.9 cd	54.1 ± 5.4 b
10	8	66.2 ± 3.9 c	45.7 ± 3.7 cd	51.1 ± 7.4 bc
10	8	65.5 ± 5.2 c	46.6 ± 2.4 cd	22.7 ± 2.8 d
16	4	81.4 ± 0.6 b	45.1 ± 2.3 cd	28.7 ± 1.0 d
16	12	83.6 ± 3.2 b	53.0 ± 3.3 b	72.8 ± 8.2 a
18.48	8	88.2 ± 0.5 a	62.7 ±2.0 a	57.3 ± 4.2 b
Positive control		Orlistat 100 ppm 78.8%	Acarbose 1000 ppm 51.9%	Allopurinol 100 ppm 78.2%

Data are expressed as the mean ± standard deviation of triplicate experiments. The mean values with different letters (a–g) in each column are significantly different (*p* < 0.05 by Duncan test).

**Table 5 foods-11-00018-t005:** Pearson correlation coefficients between flavonoid yields and enzyme inhibitory activities in subcritical water hydrolysates from immature *Citrus unshiu* pomace.

Flavonoids	Pancreatic Lipase	α-Glucosidase	Xanthine Oxidase
Hesperidin	−0.691 *	−0.799 *	−0.785 *
Hesperetin-7-*O*-glucoside (H7G)	0.551	0.635	0.379
Hesperetin (HT)	0.675 *	0.973 *	0.742 *
Narirutin	−0.653	−0.858 *	−0.820 *
Prunin (PR)	0.537	0.365	0.185
Naringenin (NG)	0.721 *	0.966 *	0.777 *
Monoglucosides (H7G + PR)	0.553	0.589	0.345
Aglycons (HT + NG)	0.686 *	0.973 *	0.750 *
Total hydrolysis products	0.691 *	0.607	0.607
% Total hydrolysis yield	0.705 *	0.654	0.654

** p* < 0.05. H7G: hesperetin-7-*O*-glycoside; PR: prunin; HT: hesperetin; NG: naringenin.

## Data Availability

The datasets generated for this study are available on request to the corresponding author.

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
