# Peer review of "Organic Acid-Catalyzed Subcritical Water Hydrolysis of Immature Citrus unshiu Pomace"

_foods, 2021, doi:10.3390/foods11010018_

Round 1
Reviewer 1 Report
In this work, the authors describe an organic acid-based subcritical hydrolysis of Immature Citrus unshiu pomace to produce flavonoid monoglucosides and aglycons with high biological activities. The concentration of citric acid and the hydrolysis period were evaluated to improve the hydrolysis yield. Moreover, the enzyme-inhibiting effects of SW hydrolysates was addressed. The document is interesting and is well structured. However, some details need to be clarified and improved. I recommend minor corrections.
Abstract
- why is important to produce flavonoid monoglucosides and aglycons with enhanced enzyme activity? Where both groups of compounds can be applied? This has to be clear to demonstrate the importance of this work and this hydrolysis.
- Line 12: “the monoglucoside yield increased” why? this information is only clear in the results. Add the % of citric acid ranges
Introduction
- Lines 72/ 73: Is this mitigation an advantage to the hydrolysis process?
- Line 83: Explain the selection of these enzymes to demonstrate the enzyme-inhibiting effects of the monoglucosides and aglycons extracted
Material and Methods
- Sample Preparation: The protocol applied is based in literature reports? Why 90 seconds? Why a powder with 14-50 mesh?
- Chemicals: The chemicals list is not so exhausting. Therefore, the chemicals must be included in the manuscript
- Hydrolysis of Acidified SW: The protocol has no references. Is it described for the first time? If not, please list the references
- Line 117: Describe why or add references to support the selection of the wt% of citric acid, the periods of time and temperature of extraction.
- High-Performance Liquid Chromatography: Did the HPLC method was designed and optimized to quantify these compounds or is based in any reported protocol?
- Line 145: Correct twofold to two-fold
- Line 153: Is Proença, not Proenca.
- Please correct the references list and verify is all the references are corrected.
- Line 167: Why 50% of methanol?
- Line 172: SPSS is Statistical Package for the Social Sciences. Add the name before the acronym.
Results and Discussion
- The discussion starts with the description of the best conditions of extractions. This paragraph must be moved to the end of this subsection and first the different conditions investigated must be detailed.
- Lines 182/183: explain better and complete this discussion. Why the pKa influences the extraction yield of the different compounds
- Line 198: why 10% of citric acid to study the effect of hydrolysis duration?
- Lines 341 to 343: These conditions lead to a predicted maximum yield. Were these conditions applied and the experimental values compared with the predicted ones? If not, these assays have to be performed. Moreover, these values need to be compared with other methods reported to demonstrate the benefits of using organic acid-based hydrolysis
- Line 349: Correct twofold to two-fold
- Table 4: The Enzyme inhibitory activities were measured after the hydrolysis using different concentration of citric acid and different periods of time. Explain the selection of this concentrations and periods. Why this enzyme inhibitory activity was not measured in the best conditions of hydrolysis (15.91% citric acid and 8.62h)?
- Line 379: compare the values reported with the ones obtained in this work. xx vs yy
- The same for the other two works reported
Conclusions
- Line 406: might be? Is this process cost-effective or not?

Author Response
Reviewer 1:
In this work, the authors describe an organic acid-based subcritical hydrolysis of Immature Citrus unshiu pomace to produce flavonoid monoglucosides and aglycons with high biological activities. The concentration of citric acid and the hydrolysis period were evaluated to improve the hydrolysis yield. Moreover, the enzyme-inhibiting effects of SW hydrolysates was addressed. The document is interesting and is well structured. However, some details need to be clarified and improved. I recommend minor corrections.
*** I really appreciate for your invaluable comments.
Abstract
1. why is important to produce flavonoid monoglucosides and aglycons with enhanced enzyme activity? Where both groups of compounds can be applied? This has to be clear to demonstrate the importance of this work and this hydrolysis.
Ans)
The following information has been added.
Line 79: to make functional foods targeting for weight control.
2. Line 12: “the monoglucoside yield increased” why? this information is only clear in the results. Add the % of citric acid ranges.
Ans)
Was revised.
Lines 11-12: (from 0 to 7% citric acid), then remained constant (from 7 to 16% citric acid), and finally de-creased (with ≥ 19% citric acid),
Introduction
3. Lines 72/ 73: Is this mitigation an advantage to the hydrolysis process?
Ans) Was revised.
Line 72: control the degradation of arabinose
4. Line 83: Explain the selection of these enzymes to demonstrate the enzyme-inhibiting effects of the monoglucosides and aglycons extracted
Ans) The following information has been added.
Line 79: to make functional foods targeting for weight control.
Material and Methods
5. Sample Preparation: The protocol applied is based in literature reports? Why 90 seconds? Why a powder with 14-50 mesh?
Ans) The powder with 14-50 mesh used was based on the general size of sample normally being used in our equipment. However, this point can be taken into consideration for future parameter analysis.
The quaternized fruit was completely crushed after 90 seconds with a blender.
6. Chemicals: The chemicals list is not so exhausting. Therefore, the chemicals must be included in the manuscript
Ans) Chemicals in Supplementary Materials has been inserted into the main text as advised (Lines 93-101).
7. Hydrolysis of Acidified SW: The protocol has no references. Is it described for the first time? If not, please list the references
Ans) Hydrolysis of acidified SW has been described for the first time.
8. Line 117: Describe why or add references to support the selection of the wt% of citric acid, the periods of time and temperature of extraction.
Ans) The ranges of the wt% of citric acid and the periods of time at fixed temperature were selected based on trial-and-error preliminary experiments.
9. High-Performance Liquid Chromatography: Did the HPLC method was designed and optimized to quantify these compounds or is based in any reported protocol?
Ans) The HPLC method was designed and optimized to quantify these compounds based on the reference [8]. (line 142) Kim, D.-S.; Lim, S.-B. Semi-continuous subcritical water extraction of flavonoids from Citrus unshiu peel: Their antioxidant and enzyme inhibitory activities. Antioxidants, 2020, 9, 360.
10. Line 145: Correct twofold to two-fold
Ans) Was revised.
11. Line 153: Is Proença, not Proenca.
Ans) Was revised.
12. Please correct the references list and verify is all the references are corrected.
Ans) I checked all the references.
13. Line 167: Why 50% of methanol?
Ans) Because methanol-diluted hydrolysates (two-fold dilution) were used for enzyme inhibitory activity analysis (lines 154-155).
14. Line 172: SPSS is Statistical Package for the Social Sciences. Add the name before the acronym.
Ans) Was revised.
Results and Discussion
15. The discussion starts with the description of the best conditions of extractions. This paragraph must be moved to the end of this subsection and first the different conditions investigated must be detailed.
Ans) Lines 246-251 were move to the end of this subsection.
16. Lines 182/183: explain better and complete this discussion. Why the pKa influences the extraction yield of the different compounds
Ans) The following information has been added.
Lines 251-253: difference in acidity (pKa values) among the organic acids [9,28]. Yu and Bulone [9] also reported that the acidity of deep eutectic solvents enhanced de-glycosylation of quercetin derivatives during extraction from apple pomace.
17. Line 198: why 10% of citric acid to study the effect of hydrolysis duration?
Ans) The 10% of citric acid was the middle value out of 4, 7, 10, 13, 16, and 19%.
18. Lines 341 to 343: These conditions lead to a predicted maximum yield. Were these conditions applied and the experimental values compared with the predicted ones? If not, these assays have to be performed. Moreover, these values need to be compared with other methods reported to demonstrate the benefits of using organic acid-based hydrolysis.
Ans) Actually, I have to do experiment at optimal hydrolysis conditions and validate the predicted value. However, I did not. I am very sorry about that.
The following has been added in the conclusion section.
Lines 429-432: Further works are needed on the validation of the predicted values by the experimental values at the optimal hydrolysis conditions for the production of flavonoid monoglucosides and aglycons, and on the measurement of enzyme inhibitory activities of the hydrolysates.
19. Line 349: Correct twofold to two-fold
Ans) Was revised.
20. Table 4: The Enzyme inhibitory activities were measured after the hydrolysis using different concentration of citric acid and different periods of time. Explain the selection of this concentrations and periods. Why this enzyme inhibitory activity was not measured in the best conditions of hydrolysis (15.91% citric acid and 8.62h)?
Ans) The enzyme inhibitory activities were measured in all the hydrolysates obtained in RSM experiments. I did not do experiment at the predicted best conditions of hydrolysis. I am very sorry about that.
The following has been added in the conclusion section.
Lines 429-432: Further works are needed on the validation of the predicted values by the experimental values at the optimal hydrolysis conditions for the production of flavonoid monoglucosides and aglycons, and on the measurement of enzyme inhibitory activities of the hydrolysates.
21. Line 379: compare the values reported with the ones obtained in this work. xx vs yy.
Ans) Has been modified.
Lines 398-400: Kim and Lim [8] measured the inhibitory activities of SW extracts from C. unshiu peel and found that the extracts at 175 °C with high in flavonoid monoglucosides and aglycons exhibited inhibitory activities of 86.8, 55.5, and 52.6% against PL, αG, and XO, respectively.
22. The same for the other two works reported.
Ans) Has been modified.
Lines 401-403: Cvetanovic et al. [36] reported that SW extract from Sambucus ebulus leaves showed high in total phenolics (116.3 mg CAE/g extract) and exhibited strong ?G inhibiting effects (2.04 mmol acarbose equivalents/g extract).
Conclusions
23. Line 406: might be? Is this process cost-effective or not?
Ans) This is process cost-effective.
Line 424 has been changed from” might” to “should”.
Reviewer 2 Report
This study describes the potential of the citric acid-catalyzed subcritical water hydrolysis for the extraction of monoglucosides and aglycons from immature Citrus unshiu. I consider that the content of the manuscript can be of interest for the readers of FOODS. However, some aspect of the manuscript must be improved.
Major points:
- Abstract: the theoretical values of citric acid concentration and hydrolysis time calculated according to the RSM are not coincident with those reported on page 8 (lines 327-331) and Table 3. Please check.
- Section 2.6: several parameters of the analytical method should be incorporated into the manuscript, such as the injection volume and the column equilibration time post-run between injections.
- Page 4, lines 188-190: authors say that “With increasing citric acid concentration, the yield of flavonoid monoglucosides (hesperetin-7-O-glycoside and prunin) first increased (from 0 to 7% citric acid), then remained constant (from 7 to 13% citric acid), and finally decreased (with > 13% citric acid)”. However, this statement is not supported by Figure 1. According to this Figure and the statistical analysis developed, the hydrolysis yield of H7G remained constant from 7 to 19 % and the PR yield from 7 to 16 %.
- Page 6, lines 283-287: please check values according to Table 1 and correct as correspond.
- Results and discussion section: I miss a further discussion of the data, for instance, comparing the extraction yield data achieved through the citric acid-catalyzed subcritical water hydrolysis with those achieved by the other published hydrolysis methods.
- Conclusions section: in this section, authors only list the same data given in the abstract and Results and discussion section. This section should be improved.
Minor comments:
- Page 2, line 61: please write “Valdivieso Ramirez et al. [27] also reported...”
- Page 3, line 117: please check punctuation.
- Page 9, lines 361: please write “PL increased 2.2-fold (from 37.0 to 81.4%)”
- Table 5: please write Hesperidin in the first line of the table
Author Response
Reviewer 2:
This study describes the potential of the citric acid-catalyzed subcritical water hydrolysis for the extraction of monoglucosides and aglycons from immature Citrus unshiu. I consider that the content of the manuscript can be of interest for the readers of FOODS. However, some aspect of the manuscript must be improved.
*** I really appreciate for your invaluable comments.
Major points:
- Abstract: the theoretical values of citric acid concentration and hydrolysis time calculated according to the RSM are not coincident with those reported on page 8 (lines 327-331) and Table 3. Please check.
Ans) Has been revised.
Lines 14-16: a citric acid concentration and hydrolysis duration of 13.34 % and 7.94 h were predicted to produce the highest monoglucoside yield of 15.41 mg/g, while 18.48% citric acid and a 9.65 h hydrolysis duration produced the highest aglycon yield of 10.00 mg/g.
- Section 2.6: several parameters of the analytical method should be incorporated into the manuscript, such as the injection volume and the column equilibration time post-run between injections.
Ans) The following information has been added
Lines 147-148: The column temperature, injection volume, and equilibrium time were 35 °C, 10 μL, and 2 min, respectively.
- Page 4, lines 188-190: authors say that “With increasing citric acid concentration, the yield of flavonoid monoglucosides (hesperetin-7-O-glycoside and prunin) first increased (from 0 to 7% citric acid), then remained constant (from 7 to 13% citric acid), and finally decreased (with > 13% citric acid)”. However, this statement is not supported by Figure 1. According to this Figure and the statistical analysis developed, the hydrolysis yield of H7G remained constant from 7 to 19 % and the PR yield from 7 to 16 %.
Ans) Has been revised.
Lines 194-195: then remained constant (from 7 to 16% citric acid), and finally decreased (with ≥ 19% citric acid).
- Page 6, lines 283-287: please check values according to Table 1 and correct as correspond.
Ans) Has been revised.
Lines 295-296: The hydrolysis yield ranged from 6.47–12.09 mg/g sample for hesperetin-7-O-glucoside, 2.32–8.05 mg/g sample for hesperetin, 2.05–3.45 mg/g sample for prunin,
- Results and discussion section: I miss a further discussion of the data, for instance, comparing the extraction yield data achieved through the citric acid-catalyzed subcritical water hydrolysis with those achieved by the other published hydrolysis methods.
Ans) The following information has been added.
Lines 308-318: Ko et al. [34] reported that aglycons such as hesperetin from lemons (about 25 mg/g sample at 190 °C) and naringenin from grapefruits (about 2.37 mg/g sample at 170 °C) were extracted at higher temperatures than glycoside such as hesperidin and narirutin because glycosides are unstable at high temperatures and long extraction times in the SW extraction and degraded to low-molecular weight aglycons at temperatures >170 °C due to their thermal instability. Ko et al. [35] indicated that naringenin (about 7.2 mg/g sample at 190 °C) could be extracted at higher temperatures (≥ 170 °C) than hesperidin and narirutin on pilot-scale SW extraction from C. unshiu peel. Kim and Lim [8] found that the SW extract from C. unshiu peel at 175 °C were high in flavonoid monoglucosides (hesperetin-7-O-glucoside 2.04 and prunin 0.31 mg/g sample) and aglycons (hesperetin 1.80 and naringenin 0.26 mg/g sample).
- Conclusions section: in this section, authors only list the same data given in the abstract and Results and discussion section. This section should be improved.
Ans) The same data given in the abstract and Results were removed in this discussion section, and the below sentence has been added.
Lines 427-429: Flavonoid monoglucosides and aglycons-rich hydrolysates could be used as a functional ingredient in the functional food and nutraceutical industries.
Minor comments:
Page 2, line 61: please write “Valdivieso Ramirez et al. [27] also reported...”
Ans) Was revised.
Page 3, line 117: please check punctuation.
Ans) Was revised.
Page 9, lines 361: please write “PL increased 2.2-fold (from 37.0 to 81.4%)”
Ans) Was revised.
Table 5: please write Hesperidin in the first line of the table
Ans) Was revised.
Round 2
Reviewer 2 Report
The manuscript was improved with respect to the previous version. However, some aspects are not fully corrected.
- Section 2.6: based on my expertise, a column equilibration time of 2 min between injections is not enough to stabilize the column in the initial conditions of the mobile phase, especially considering the column size (250 mm) and flow rate used in this work. Under these conditions, equilibration times higher than 10 min are typically required to obtain good retention times reproducibility among injections.
- Page 5, lines 192-195: authors say that “With increasing citric acid concentration, the yield of flavonoid monoglucosides (hesperetin-7-O-glycoside and prunin) first increased (from 0 to 7% citric acid), then remained constant (from 7 to 16% citric acid), and finally decreased (with ≥ 19% citric acid)”. However, this is incorrect since citric acid concentrations higher than 19 % were not assayed. This text should be rewritten as follows: “With increasing citric acid concentration, the yield of flavonoid monoglucosides (hesperetin-7-O-glycoside and prunin) increased from 0 to 7% citric acid. Afterward, the hesperetin-7-O-glycoside yield remained constant (from 7 to 19% citric acid) while the pruning yield decreased with 19% of citric acid”. In addition, this aspect should also be corrected in the abstract.
- Page 7, line 295: please write “1.99–8.05 mg/g sample for hesperetin”.
Author Response
I would like to express my deepest gratitude to the reviewer for the appreciation and valuable comments. Revised portions are conducted by using “Tracked Changes” and red fonts are the revision that have been made.
Response for Reviewer’s Comments:
The manuscript was improved with respect to the previous version. However, some aspects are not fully corrected.
-Section 2.6: based on my expertise, a column equilibration time of 2 min between injections is not enough to stabilize the column in the initial conditions of the mobile phase, especially considering the column size (250 mm) and flow rate used in this work. Under these conditions, equilibration times higher than 10 min are typically required to obtain good retention times reproducibility among injections.
Ans) Thank you for your comments. This point will be taken into consideration for future HPLC analysis.
- Page 5, lines 192-195: authors say that “With increasing citric acid concentration, the yield of flavonoid monoglucosides (hesperetin-7-O-glycoside and prunin) first increased (from 0 to 7% citric acid), then remained constant (from 7 to 16% citric acid), and finally decreased (with ≥ 19% citric acid)”. However, this is incorrect since citric acid concentrations higher than 19 % were not assayed. This text should be rewritten as follows: “With increasing citric acid concentration, the yield of flavonoid monoglucosides (hesperetin-7-O-glycoside and prunin) increased from 0 to 7% citric acid. Afterward, the hesperetin-7-O-glycoside yield remained constant (from 7 to 19% citric acid) while the pruning yield decreased with 19% of citric acid”. In addition, this aspect should also be corrected in the abstract.
Ans) Was revised.
Lines 193-196: With increasing citric acid concentration, the yield of flavonoid monoglucosides (hesperetin-7-O-glycoside and prunin) increased from 0 to 7% citric acid. Afterward, the hesperetin-7-O-glycoside yield remained constant (from 7 to 19% citric acid) while the pruning yield decreased with 19% of citric acid.
Lines 10-13: with 8 h of hydrolysis and an increasing citric acid concentration, the yield of flavonoid monoglucosides (hesperetin-7-O-glycoside and prunin) increased from 0 to 7% citric acid. Afterward, the hesperetin-7-O-glycoside yield remained constant (from 7 to 19% citric acid) while the pruning yield decreased with 19% of citric acid,
- Page 7, line 295: please write “1.99–8.05 mg/g sample for hesperetin”.
Ans) Was revised (line 297).